# Management Assessment and Future Projections of Construction and Demolition Waste Generation in Hai Phong City, Vietnam

**Kien Ton Tong** [1,2], **Ngoc Tan Nguyen** [2,3], **Giang Hoang Nguyen** [2,3,*], **Tomonori Ishigaki** [4] **and Ken Kawamoto** [2,5]

1. Building Materials Faculty, Hanoi University of Civil Engineering, No. 55 Giai Phong Street, Hai Ba Trung District, Hanoi 100000, Vietnam
2. Innovative Solid Waste Solutions (Waso), Hanoi University of Civil Engineering, No. 55 Giai Phong Road, Hai Ba Trung District, Hanoi 100000, Vietnam
3. Faculty of Building and Industrial Construction, Hanoi University of Civil Engineering, No. 55 Giai Phong Street, Hai Ba Trung District, Hanoi 100000, Vietnam
4. Material Cycles Division, National Institute for Environmental Studies, 16-2 Onogawa, Tsukuba 305-8506, Japan
5. Graduate School of Science and Engineering, Saitama University, 255 Shimo-okubo, Sakura-ku, Saitama 338-8570, Japan
* Correspondence: giangnh@huce.edu.vn; Tel.: +84-932247588

**Abstract:** Along with economic development, urbanization will generate a large amount of solid waste and put pressure on the waste management systems in developing countries. Face-to-face interview methods were used to investigate the current status of construction and demolition waste (CDW) management (collection, transportation, treatment, and storage) as well as reveal attitudes of governmental agencies and enterprises towards CDW recycling and recycled material products in Hai Phong City, Vietnam. Waste generation rates (WGRs) of the works were also determined by site surveys and as-built drawings method of typical old buildings to be demolished and two licensed new construction works. WGRs of 34.5 kg/m$^2$ and 758 kg/m$^2$ were identified during the construction and demolition of small private houses, respectively, while WGRs at public house demolition sites were 1053 kg/m$^2$. To effectively manage the CDW, the gross floor area of new construction work was estimated by a multiple regression equation with the population and gross region domestic product growth. Based on this model combining the investigation results, the amount of CDW increase in 5–30 years is also predicted. This data set will help management agencies plan storage yards as well as select the appropriate CDW treatment and recycling methods, contributing to building a sustainable and effective CDW management model for Hai Phong City as well as Vietnam in the future.

**Keywords:** construction and demolition waste (CDW); CDW dumping site; waste generation rate (WGR); projection of CDW generation; Hai Phong City

## 1. Introduction

The determination of the CDW generation rate is very important not only in establishing the current data set, but also for future forecasts, to meet the sustainable CDW management [1]. To date, five widely used methods to determine and forecast CDW generation are (i) site visit, (ii) generation rate calculation (GRC), (iii) lifetime analysis, (iv) classification system accumulation, and (v) variable modeling [2]. For regions/localities in which the primary data is not available, site visits and GRC can be adopted by examining either the as-built drawings or interviewing construction contractors. These methods predict the CDW generation rate from a target site, i.e., waste generation rate (WGR), as well as information on waste collection and transportation, temporary storage yards, and

final disposal. WGRs of CDW as well as total solid waste have been intensively studied in many countries [3–9], and the proportion of CDW generated to total solid waste is linked to the waste generation per capita and/or the GDP of the region/country. Llatas et al. [8], for example, showed that CDW accounted for 82.7% of total solid waste generated by economic activities and 48% of total solid waste in most EU countries. Majlessi et al. [9] showed that CDW was 84% of the total solid waste and the CDW generation rate could be predicted by a quadratic regression model using the building permits. In addition, the composition and percent of material purchased for building materials and mass/volume of materials per gross floor area ($kg/m^2$ or $m^3/m^2$) are used to estimate WGRs [10–16].

To forecast future CDW generation, several estimation models have been proposed, including an empirically-based regression model [17], a model and planning tool based on the building information modeling system [18], and computer predictive modelling through gene expression programming [19]. Islam et al. also studied the prediction of CDW generation by regression analysis [20]. Recently, Qiao et al. [1] estimated the building area to predict the amount of CDW waste generated in Shandong Province of China, and the quadratic exponential smoothing method was used for forecasting future waste generation. The Ministry of Land, Infrastructure, Transport and Tourism (MLIT) of Japan [21] is adopting sensitivity analyses considering multiple factors such as population growth, economic growth, WGR and gross floor area, and percentage of demolished/renovated areas to determine the range of future projection of CDW generation. However, the amount of CDW generated depends not only on the building floor area but also on the economic growth, degree of urbanization, population, and other factors [22]. Besides, it is important to suggest a suitable and easy applicable forecasting model fully considering the access to the statistical data and available information on a target region/country.

Vietnam has the high economic growth and rapid urbanization, especially in the big cities. The average annual economic growth rate is 6.0% in 2015–2020, and the urbanization rate increased rapidly from 35.7% in 2015 to 39.3% in 2020 and is expected to be approximately 45% in 2025 [23]. The fastest urban growth rate was observed in Ho Chi Minh City (approximately 64.2%), and the urban growth rates of other big cities such as Can Tho City, Hanoi City, Da Nang City, and Hai Phong City also ranged between 46.7 and 60% [24]. These lead to the high amount of CDW generated annually, with 10–15% of total urban solid waste in Vietnam. In specific urban areas such as Hanoi and Ho Chi Minh Cities, especially, the amount of CDW generated reaches 25% [25]. But the main treatment of CDW generated is landfilling (85–90%), and the rate of reuse and recycling has been reported approximately 10% [10] and only 1–2%, respectively [26]. However, Vietnam still does not have a complete construction and demolition waste (CDW) management system due to a shortage of policies and law enforcement in solid waste management. Vietnam faces serious difficulty in promoting CDW recycling in the construction industry and in finding new sites for CDW disposal. In order to promote the reuse and recycling of CDW, the Vietnamese Ministry of Construction (MOC) issued Circular No. 08/2017/TT-BXD on CDW management [27]. This circular prescribes the classification, collection, reuse, recycling, and treatment of solid construction waste in Decree No. 38/2015/ND-CP [28]. However, most enterprises, including waste generators and government agencies responsible for CDW management, have not paid much attention to implementation of Circular No. 08/2017/TT-BXD [29]. Thus, it is urgent to develop a sustainable waste management system applying advanced CDW treatment technologies, as well as to reduce CDW generation at the source and segregate the generated CDW on-site at the source (i.e., newly built and demolished old buildings and construction sites).

In Vietnam, studies of WGR and CDW and future projections of CDW generation are limited. Hoang et al. [29] conducted a review of a wide array of documents related to CDW management in Southeast Asia (SEA). The definitions, the current state of CDW generation and composition, flows, and institutional arrangements for CDW management were assessed. This study suggested a need for a more holistic and aggressive method of sustainable CDW management, namely developing legalized systematic approaches

to CDW data collection and database establishment. Moreover, quantifying CDW generation is also limited to large cities like Hanoi in Vietnam [10,15,16]. Hoang et al. [10] surveyed a series of 15 building construction and demolition sites in Hanoi, Vietnam, to identify waste generation rates (WGRs), composition, and current handling practices of CDW. Nghiem et al. [16] conducted an in-depth investigation of various aspects of CDW generation and management in Vietnam. Their insights provide valuable information on the current situation, practices, and attitudes towards CDW recycling, but these studies concentrated on the WGRs of several main materials generated from old building demolition sites in Hanoi. To implement Directive 41/TTG dated December 2020 on urgent solutions to strengthen solid waste management [30], as well as contribute to the realization of the goals of the National Strategy for solid waste management in 2025 [31] (to achieve 90% collection and 60% recycling of CDW), an applicable estimation method to forecast the CDW generation that suits the current CDW status in Vietnam is required.

Among the big cities in Vietnam, Hai Phong has a relatively high economic growth rate and is the second-largest city in northern Vietnam. Hai Phong City had an average annual gross regional domestic product (GRDP) growth rate of 13.94% in 2016–2020, which is 1.97 times higher than that in 2011–2015 [32]. According to the population and housing census in 2019 [33], Hai Phong City has 2,028,514 residents (7th largest city nationwide), and the population density is 1299 people/km$^2$ (4.5 times higher than the national average of population density in Vietnam). The construction industry and service still play an important role in contributing to the economic development of the city. These sectors increased from 92.48% in 2015 to 95.08% GRDP in 2020. The construction industry accounts for 42.3–49.7% of all industries in Hai Phong City [34]. Due to the rapid development and urbanization, Hai Phong City generated approximately 93,440 tons of CDW annually, reaching 12–13% of the total amount of solid waste produced in 2015 [35]. This amount of CDW is forecasted to increase to 339,450 tons/year by 2030 [36]. In order to promote sustainable and sound solid waste management including CDW, the city has promulgated many edicts and related policies on solid waste management (see Appendix A). According to the city master plan of Hai Phong City [37], for example, seven domestic solid waste treatment zones, seven district treatment zones, and five landfills for CDW disposal, namely Trang Cat (5 ha), Ben Gung (3 ha), Dong Hoa (3 ha), Do Son (3 ha), and Lai Cach (3 ha), are planned. In reality, however, Hai Phong City is facing severe environmental and social problems due to the insufficient treatment of solid waste collected and stored at landfill sites, suggesting that it is important to identify the current CDW generation to sketch an overall picture of awareness and difficulties in CDW recycling and to project the future CDW volume in Hai Phong City.

Based on the above facts, this study used survey methods by questionnaire interview and field survey to investigate the current status of collection, transportation, treatment, and storage of CDW as well as reveal the attitudes of governmental agencies and enterprises towards CDW recycling in Hai Phong City. To determine the WGRs of both construction and demolition works, both site surveys and as-built drawings were used. Based on the statistic data on population and GRDP of Hai Phong city in 1999–2020 combined with actual area of floors of residential buildings constructed in 2010–2020, a multivariate regression analysis model was proposed to forecast the total floor area of houses built and the amount of CDW generated in the next 5–30 years. These results will help management agencies plan storage yards as well as select the appropriate CDW treatment and recycling methods, contributing to building a sustainable and effective CDW management model for Hai Phong City as well as Vietnam.

## 2. Methodologies

### 2.1. Data Collection and Interview Survey with Local Authorities and Construction Enterprises

To reveal the attitudes of governmental agencies and enterprises towards CDW recycling and management, as well as towards recycled material products, structured questionnaires and face-to-face interviews were conducted with agencies/enterprises in all districts

of Hai Phong City (Table 1). Two groups were interviewed consisting of (i) the local authorities in charge of CDW management, such as the Haiphong Department of Construction (DOC); Department of Natural Resources and Environment (DONRE); Natural Resources and Environment Bureau (NREB) of 15 districts; Project Management Unit (PMU) or Urban Management Bureau (UMB) of 7 urban districts, and Economy-Infrastructure Bureau (EIB) of 8 rural districts; and (ii) a group of construction enterprises including five construction contractors and five demolition contractors. Face-to-face meetings were arranged by e-mail and telephone, and the survey form was sent via e-mail at least five working days before the interview so that agencies had time to study its content.

### 2.2. Field Survey of Old and New Buildings

From the information collected by the Hai Phong DOC, the research team planned a site visit to new construction sites and demolition sites to determine the CDW generation rates (i.e., WGR). On-site measurement, development of drawings of old buildings, and analysis of as-built drawings of new buildings were performed to calculate and predict the amounts of CDW generated from old building demolition works in the coming years. Information collected on old buildings included the name, address, year of construction, number of floors, gross floor area (GFA), and type of structure. These buildings were then categorized according to the construction period (e.g., in the 1950s, '60s, '70s, '80s, and '90s) to predict the CDW amounts that will be generated during their demolition at the end of their service life in the next 5, 10, 20, and 30 years. For new buildings, a list of buildings newly permitted in all districts in the five last years (2016–2020) and their typical as-built drawings were obtained to predict the CDW amounts that will be generated during their demolition in the next 30–60 years. For both old and new buildings, on-site observations and interviews were conducted to verify the results of the as-built drawing analysis. The location of districts in Hai Phong City and all survey sites are shown in Figure 1.

### 2.3. Field Survey at CDW Landfill/Dumping Sites

In order to survey the current status of CDW dumps, this study used check sheets during the site survey at both official and illegal dumpsites [38]. The main contents of the survey include general information about location, years of operation, management unit, and detailed information about the type and amount of waste dumped over time and treatment methods at the yard.

**Table 1.** List of contacted agencies and enterprises for interview survey in Hai Phong City.

| No. | City/District/Enterprises | Agency | No. of Interviewers | Items of Collected Data and Questionnaires |
|---|---|---|---|---|
| | | | | City level |
| 1 | | DOC | 3 | • Total number of old buildings to be demolished and new construction permissions in 2016–2020<br>• Old buildings built in the period of 1960 to 1990 (name/address, year of inauguration, scale/number of floors, gross floor area listed in Decree No. 101/2015/ND-CP [39] (1); Status of implementation of Decision No. 1711/2012/UBND<br>• One set of as-built drawings of a demolished building and two sets of drawings of newly constructed buildings |
| | Hai Phong City | | | |
| 2 | | DONRE | 2 | • Actual state of solid waste collection, transportation and treatment in Hai Phong City<br>• Integrated solid waste management plan to 2025, vision to 2050<br>• Forecast of solid waste generation in the city |
| | | | | District level |
| | Urban districts | | | A. General information<br>B. Content of survey |
| 1 | Duong Kinh | UMB, PMU | 2 | 1. Current situation of generated solid waste in the local region: Year, Total amount of MSW (tons/year), % of CDW to total MSW, Collection % of CDW, Recycled % of CDW |
| 2 | Do Son | NREB | 1 | |
| 3 | Hai An | UMB | 1 | 2. Level of compliance/implementation of the regulation on construction solid waste management (Circular No. 08/2017/TT-BXD [27]) (Select: High, Medium, Low, Not applicable) |
| 4 | Kien An | UMB, PMU | 2 | |
| 5 | Hong Bang | NREB | 1 | 3. Treatment and recycling methods of CDW: Backfill at planned landfills, Backfill at illegal dumping sites, Reuse for ground leveling, Recycled to make construction and/or other materials, others (select multiple answers) |
| 6 | Ngo Quyen | NREB | 1 | |
| 7 | Le Chan | UMB, PMU | 2 | 4. Information of CDW landfills: Name/address, Area (ha), Start of Service (year), End of Service (year), Notes |
| | Rural districts | | | 5. Information of local companies/contractors that demolish, collect and/or transport CDW: Name and total number of companies/contractors, Scope of work, contact |
| 8 | An Duong | EIB | 1 | |
| 9 | An Lao | NREB, EIB | 2 | 6. Orientations for promoting CDW recycling, supporting policies and/or investment for recycling enterprises |
| 10 | Bach Long Vi | EIB | 1 | 7. Difficulties and obstacles in the management and recycling of CDW (select multiple answers) |
| 11 | Cat Hai | EIB | 1 | 8. Information of old and licensed construction within the local area: |
| 12 | Kien Thuy | EIB | 1 | 8.1. Newly licensed constructions in 4 years from 2017 to 2020: Permission of public construction works, Permission of urban housings, Permission of rural housings, Permission of temporary works, Total number of construction works exempted from permission, Project name/address, Year of license, Investor, Scale/number of floors, Gross floor area |
| 13 | Tien Lang | NREB | 1 | |
| 14 | Vinh Bao | NREB | 1 | |
| 15 | Thuy Nguyen | NREB, EIB | 2 | 8.2. Old buildings to be demolished from 2017 to 2020: Name/address, Year of inauguration, Scale/number of floors, Gross floor area) |

**Table 1.** *Cont.*

| No. | City/District/Enterprises | Agency | No. of Interviewers | Items of Collected Data and Questionnaires |
|---|---|---|---|---|
| | | | | Construction enterprises |
| 1 | Construction contractors | - | 5 | A. Basic Information on Company |
| | | - | | 1. Total number of employees<br>2. Main field of operation of company<br>3. Type of license of company<br>4. Assets of company (heavy machinery)<br>5. Experience in demolition works: years and works/year |
| | | | | B. Management of Demolished Waste |
| 2 | Demolition contractors | | 5 | 1. Own instructions (manual, guideline) of demolition works<br>2. Knowledge on legal documents that require contractors to classify demolition waste<br>3. Opinion to current legislations/legal framework on demolition waste management: Ease of following, Appropriate, Economically efficient, Necessary (Linguistic variables: Totally agree, Agree, Neutral, Disagree, Totally disagree)<br>4. Conditions of sorting/classification of demolished waste<br>5. Main difficulties to sort demolished waste: Cost increase, Manpower shortage, Time consuming, Lack of skill and technology, Lack of machinery, Small on-site area, Lack of legal guidelines, Lack of treatment facility post-demolition, Lack of awareness of workers (Linguistic variables: Totally agree, Agree, Neutral, Disagree, Totally disagree)<br>6. Eagerness to sort/classify demolished waste<br>7. Objectives for the sorting/classification criteria of demolished waste<br>8. Understanding of engineers and workers of the regulation of construction solid waste management (Circular No.08/2017/TT-BXD) |

DOC: Department of Construction; DONRE: Department of Natural Resources and Environment; UMB: Urban Management Bureau; PMU: Project Management Unit; NREB: Natural Resources and Environment Bureau; EIB: Economy-Infrastructure Bureau. [1] Among 205 old buildings, 73 of different ages were investigated to calculate gross floor area (GFA in $m^2$) in this study.

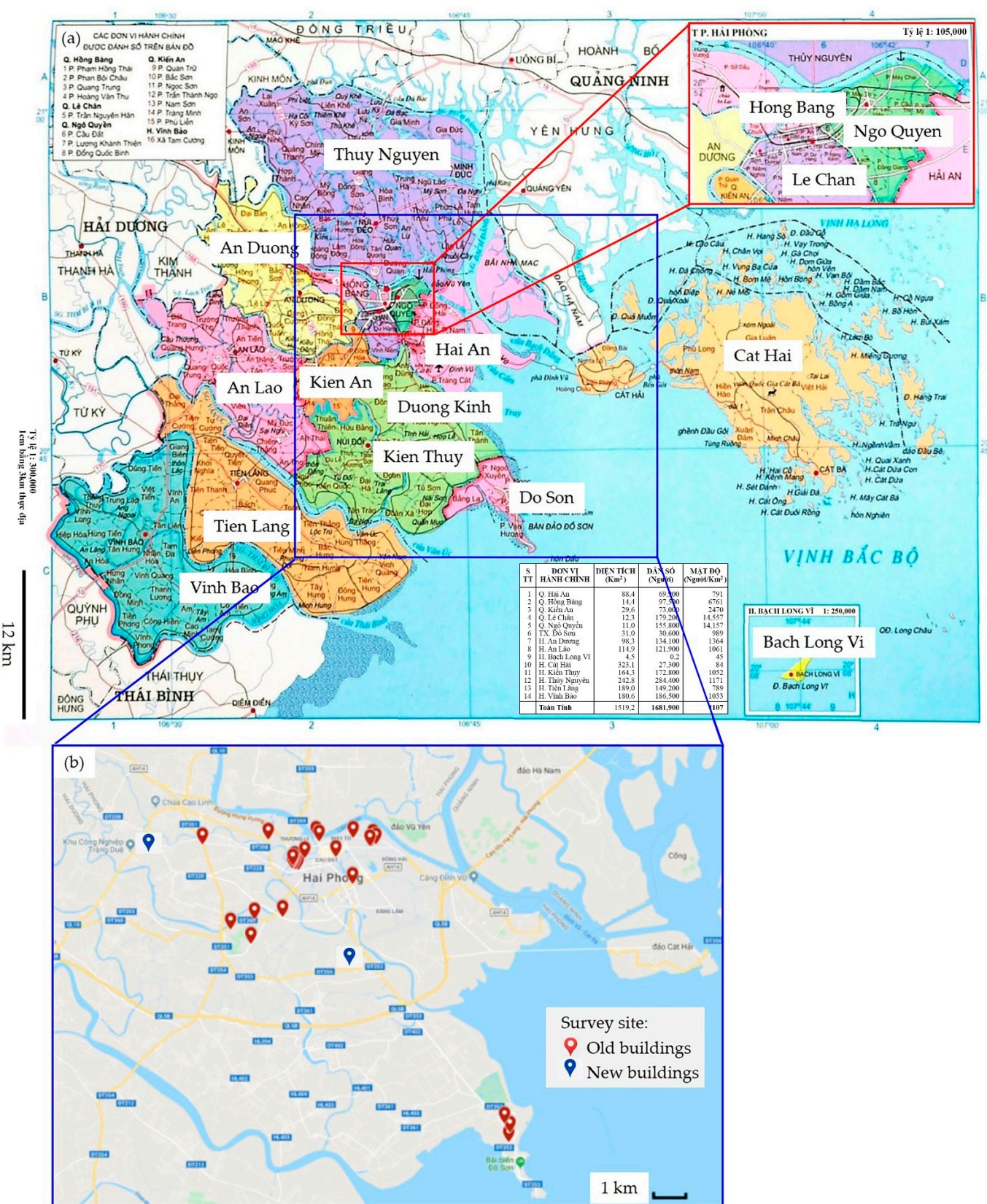

**Figure 1.** (**a**) Districts in Hai Phong City. (**b**) Survey sites of old buildings to be demolished and new buildings (only survey sites in urban districts are shown; map data © 2021 Vietnamitas en Madrid and © 2021 Google).

*2.4. Calculation of Waste Generation Rates and CDW Projection*

2.4.1. Waste Generation Rates

To determine the waste generation rates (WGRs) from old buildings to be demolished and new buildings, one set of as-built drawings of a demolished building and two sets of drawings of newly constructed buildings were studied in detail to determine the amount of CDW generated. Three main CDW categories of focus are crushed concrete (CC), crushed brick (CB), and reinforcing steel (S).

The volume of concrete ($V_C$) was determined by subtracting the volume of reinforcing steel ($V_S$) from the total volume of slabs, beams, and columns ($V_{RC}$), while that of brick ($V_B$) was determined by calculating the total volume of brick walls. By studying the quantity survey spreadsheet of demolition contractors, two values $V_{RC}$ and $V_B$ were obtained, together with an estimated mass of reinforcing steel ($m_S$). Volumes $V_S$ and $V_C$ were calculated by Equations (1) and (2):

$$V_S = \frac{m_S}{\rho_S} \left[ \text{m}^3 \right] \tag{1}$$

In Equation (1), the density of steel was taken as $\rho_S = 7850 \text{ kg/m}^3$.

$$V_C = V_{RC} - V_S \left[ \text{m}^3 \right] \tag{2}$$

This converts volumes of $V_C$ and $V_B$ to a mass by multiplying them by the bulk densities of concrete without reinforcing steel ($\rho_c$) and clay brick ($\rho_B$) by Equations (3) and (4), in which $\rho_C = 2200 \text{ kg/m}^3$ [40,41], and $\rho_B = 1500 \text{ kg/m}^3$ [1].

$$m_C = V_C \times \rho_C \tag{3}$$

$$m_B = V_B \times \rho_B \tag{4}$$

By recording the mass of steel collected for sale as scrap and the number of transport trucks from the demolition site, two values, $m_S$, and total $V_C + V_B$, can be easily obtained. The generation rate of CC+CB is then calculated per GFA by the following equation:

$$G_{CC+CB} = \frac{m_C + m_B}{GFA} \left[ \text{kg/m}^2 \right] \tag{5}$$

Employing the value of the estimated mass $m_S$ of reinforcing steel, the generation rate of steel can be determined as follows:

$$G_S = \frac{m_S}{GFA} \left[ \text{kg/m}^2 \right] \tag{6}$$

The generation rate of the main categories of CDW from a demolition site is calculated by Equation (7):

$$G_{Demolition}^{OB} = G_{CC+CB} + G_S \ \left[ \text{kg/m}^2 \right] \tag{7}$$

Because complete sets of design drawings can be more easily obtained for new buildings, a detailed quantity survey was carried out to determine the mass of concrete (without reinforcing steel) $m_C^{Con.}$, brick $m_B^{Con.}$, and reinforcing steel $m_S^{Con.}$ used for the project and also how much of this material was discarded during construction ($m_C^w$, $m_B^w$, $m_S^w$). By measuring the GFA on floor plans, the CDW generation from construction of these new buildings ($G_{Construction}^{NB}$) and demolition when they are demolished at the end of their service life ($G_{Demolition}^{NB}$) were estimated by the following equations:

$$G_{Construction}^{NB} = \frac{m_C^w + m_B^w + m_S^w}{GFA} \ \left[ \text{kg/m}^2 \right] \tag{8}$$

$$G_{Demolition}^{NB} = \frac{m_C^{Con.} + m_B^{Con.} + m_S^{Con.}}{GFA} \ \left[ \text{kg/m}^2 \right] \tag{9}$$

2.4.2. CDW Projection

To predict the generation of CDW in the future, we considered CDW arising from two sources and calculated it as follows: (i) solid waste generated from new construction is equal to the total floor area in the year of the construction multiplied by the coefficient of waste generated during the construction process; and (ii) solid waste arising from an old building scheduled to be demolished in the year. This is equal to the total floor area of old large/public buildings and old small/residence houses to be multiplied by the coefficient of waste generation due to the demolition. From the CDW generation rates during construction and demolition of the works, we can calculate the CDW data set generated in each year or the next 5, 10, 20, 30, vision to 50 years:

$$\text{Total CDW} = \text{new GFA} \times G_{Construction}^{NB} + \text{old large GFA} \times G_{Demolition}^{OLB} + \text{old small GFA} \times G_{Demolition}^{NSB} \quad (10)$$

where the emission factors are affected by the percentage of waste that is reused on-site, the level of construction and demolition technology, the proportion of waste components, and the classification and recycling of CDW on site (according to Circular 08/BXD from May 2017 requiring on-site sorting and recycling and Law No. 72/2020/QH14 effective from January 2022 on environmental protection). These additional coefficients may increase due to a reduction of construction costs and scale of works, but they will decrease due to measures to minimize the generation of by-products. Therefore, it is difficult to determine the rate of change of these coefficients in the future, and these coefficients are used with the survey results obtained in 2020.

The total newly built floor area was estimated by multiple regression with the main variables being population ($X_1$) and economic growth rate ($X_2$). The larger the population, the higher the demand for newly built residential housing. High economic growth will increase demand for office buildings and other construction. The form of the regression equation is:

$$[\text{Total new construction floor area in the future}] = a_1 \times [X_1] + a_2 \times [X_2] + b \quad (11)$$

where $X_1$ and $X_2$ are independent variables that may be population (P), gross region domestic product (GRDP), or population growth rate (P rate) and GRDP growth rate (GRDP rate). The coefficients of $a_1$, $a_2$, and b are determined from the total floor area of houses built from the year 2010 to 2020 (Table 2).

**Table 2.** Actual area of floors of residential buildings constructed (actual GFA), gross regional domestic product (GRDP), population (P), growth rate compared to the last year (GRDP rate), and population growth compared to the last year (P rate) in 2010–2020.

| Year | Actual GFA (Thsnd. m²) | GRDP (Bil. VND) | Population, P (Thsnd. Persons) | GRDP Rate (%) | P Rate (%) |
|------|------------------------|------------------|-------------------------------|----------------|-------------|
| 2010 | 1337.5 | 70,549 | 1862.9 | 11.68 | 1.22 |
| 2011 | 1617.5 | 86,916 | 1886.2 | 7.05 | 1.25 |
| 2012 | 1450.5 | 97,069 | 1912.9 | 13.93 | 1.42 |
| 2013 | 1477.5 | 103,908 | 1932.2 | 10.92 | 1.01 |
| 2014 | 1486.0 | 118,384 | 1950.7 | 13.91 | 0.96 |
| 2015 | 1328.8 | 131,314 | 1969.5 | 16.44 | 0.96 |
| 2016 | 1711.7 | 149,584 | 1985.3 | 20.73 | 0.80 |
| 2017 | 1948.6 | 174,182 | 2001.4 | 17.21 | 0.81 |
| 2018 | 2392.5 | 210,295 | 2016.4 | 11.22 | 0.75 |
| 2019 | 2411.1 | 246,485 | 2033.2 | 12.38 | 0.83 |
| 2020 | 2612.0 | 267,091 | 2053.5 | 8.36 | 1.00 |

Source: Hai Phong CSO, 2019 [33] and General Statistics Office of Vietnam.

## 3. Results and Discussion

### 3.1. Current Status of CDW Management

The results of interviews with local authorities and enterprises confirmed that insufficient attention had been paid to CDW management. As a result, data related to CDW in certain localities and the whole city have not been collected. Little attention has been paid to developing statistical data on CDW generation or the ratio of CDW to solid waste in these localities. All district officers who participated in the survey did not have all the information on the CDW ratio to municipal solid waste. Among 15 districts, only three provided the data on municipal solid waste including CDW amounts in 2018. Therefore, it is necessary to establish a database and forecasting system for CDW generation in annual as well as the short- and long-term projections [29,42]. To do so, it is essential to continue to conduct surveys and automatically collect CDW data at newly licensed construction works as well as old works being renovated and demolished throughout the city.

Regarding the implementation of CDW management policies, the results of the self-assessment of 19 official interviewees of their awareness and execution of Circular No. 08/2017/TT-BXD [27] indicated that only one interviewee self-assessed as having a high level of compliance to the Circular, making up 5% of the total pool of responses. Meanwhile, up to 4/19 (21%), 8/19 (42%), and 6/19 (32%) interviewees rated compliance at the respective levels as a medium, low, and not yet applied. The average of all numerical values yields a final value of 0.33. Referring to the scale of conversion, the overall level of implementation of Circular No. 08/2017/TT-BXD in Hai Phong can be considered low. Besides that, all surveyed contractors (10/10 contractors in the construction and demolition field) were unaware of any legal document on CDW management, including Circular No. 08/2017/TT-BXD. Nevertheless, all contractors agreed that regulations must be implemented so that CDW is managed and recycled properly.

A list of difficulties in CDW management was also summarized from 19 interview forms. The results are sorted from highest to lowest level of consensus as follows: (i) the highest is no professional enterprise in gathering CDW, and no proper planning for CDW landfill (11/19, or 57.89%); (ii) low awareness by stakeholders of the importance of treating and recycling CDW (06/19, or 31.58%); (iii) rapid rates of urbanization, together with the fact that construction and demolition projects are scattered, making it hard to gather CDW (03/19, or 15.79%); and (iv) the lack of official guidance in transporting and treating CDW and the official norms of these activities (02/19, or 10.53%). Communication to raise awareness in the community has not been carried out regularly and continuously. Therefore, the city needs to include training on CDW management for state managers and enterprises related to the construction industry to prevent and reduce the generation of CDW substances, and use recyclables and environmentally friendly materials in the construction process [43]. It is important to implement, supervise, and inspect the classification, collection, and transportation of CDW following Circular 08/TTBXD and the national strategy on integrated solid waste management by 2025, with a vision to 2050 [30,31,42].

For efficient recycling, CDW must first be sorted into different categories. This extra step may challenge construction and demolition contractors in several aspects. The survey gave all interviewees a list of nine possible subjective and objective difficulties that they would potentially face when the sorting of CDW is implemented. Each difficulty was assessed by the contractors on five levels, from virtually no challenge to the most challenging. These linguistic variables were then processed by fuzzy logic with a scale ranging from 1 (highest numerical value—the most challenging) to 9 (lowest numerical value—the least challenging). If more than one challenge had the same score, they shared a common ranking number. The top four challenges are cost increase, time consumption, lack of post-demolition treatment facilities, and manpower shortage. Among these four, the first two are subjective and linked (cos)—time). The manpower shortage is also a subjective challenge, while the lack of post-demolition treatment facilities is objective and needs special attention by policymakers to enable such conditions. In addition, the MOC is nominated to assume the prime responsibility and coordinate with the Ministry of Natural

Resources and Environment (MONRE) to urgently build and promulgate a synchronous system of standards, regulations, and economic and technical norms in the design of a collection CDW system suitable for classification at the source and centralized demolition and construction sites. For localities, it is necessary to control infrastructures for collection and transportation according to the approved construction planning. Technical guidance documents on the classification, treatment, and recycling of CDW are indispensable for promoting the usage of CDW as raw materials and minimizing landfilling. The city people committee (PC) assigns the prime responsibility for formulating a solid waste management plan and participating in CDW management to the DOC. Planning and arranging CDW gathering and transshipment points in urban centers or concentrated rural residential areas could be practical solutions to ensure environmental sanitation under regulations.

The planning of solid waste dumping sites in Hai Phong has been approved since 2012 according to Decision No. 1711/UBND [37]. The storage, collection, transportation, and treatment of CDW, however, have not met the requirements for environmental protection, which has led to the generation of common illegal dumping sites, disrupting the social order and safety. The landfills are used mainly for receiving and treating municipal solid waste. Therefore, it is essential to incorporate plans for managing CDW within the general strategy for solid waste [10]. The city PC needs to review and adjust the solid waste disposal plan to develop a centralized CDW recycling treatment facility at the city level in the approved subdivision plan, mobile recycling plants in the centralized areas, and CDW transfer yards at the district level. These projects have identified investment needs, priorities, and phased tasks of the city's master plan. The city should also issue CDW-designated regulations, guidance on, and control over the performance of service contracts for CDW collection, transportation, and treatment. Additionally, it is necessary to develop a roadmap and direct organizations to thoroughly treat illegal CDW landfills and prevent the formation of new temporary landfills.

### 3.2. Handling and Recycling Policies of CDW

The interview form gave interviewees five choices for treatment and recycling methods of CDW (if available). Several methods could be selected simultaneously. The results of this survey are displayed in Table 3 and Figure 2. The most popular treatment of CDW in Hai Phong is currently "Reuse for levelling" with 16/19 interviewees (82%) confirming the adoption of this method in their local regions. Two other methods that are also quite popular are landfilling at official or illegal dumping sites (35% and 29%, respectively). Finally, three of nineteen interviewees mentioned the reuse of undamaged clay brick recovered during demolition for new construction. However, this is hardly any surprise and is also limited to a small scale. This method pollutes the environment and affects the urban landscape while wasting resources that could be used to produce recycled materials. Therefore, together with the planning of official CDW landfills in the future and the achievement of the goal of the national integrated solid waste management strategy by the end of 2025, 90% of the total amount of CDW generated in urban zones will be collected and treated to meet environmental protection requirements, of which 60% will be reused or recycled into recycled products and materials by appropriate technologies [31]. The city needs to actively develop and issue policies to encourage and attract investment in infrastructure for collection, storage, consolidation, transportation, and CDW recycling using modern technology, simultaneously simplifying procedures for investment preparation, construction, and operation of waste treatment facilities [30]. In parallel, several proposed solutions can also be utilized: (i) actively seek and encourage all economic sectors to invest in the classification, collection, transportation, and treatment of CDW; (ii) identify and announce prices for these services in the locality; and (iii) urgently promote investment in or put into operation CDW recycling plants in the form of public-private partnerships following local socio-economic conditions.

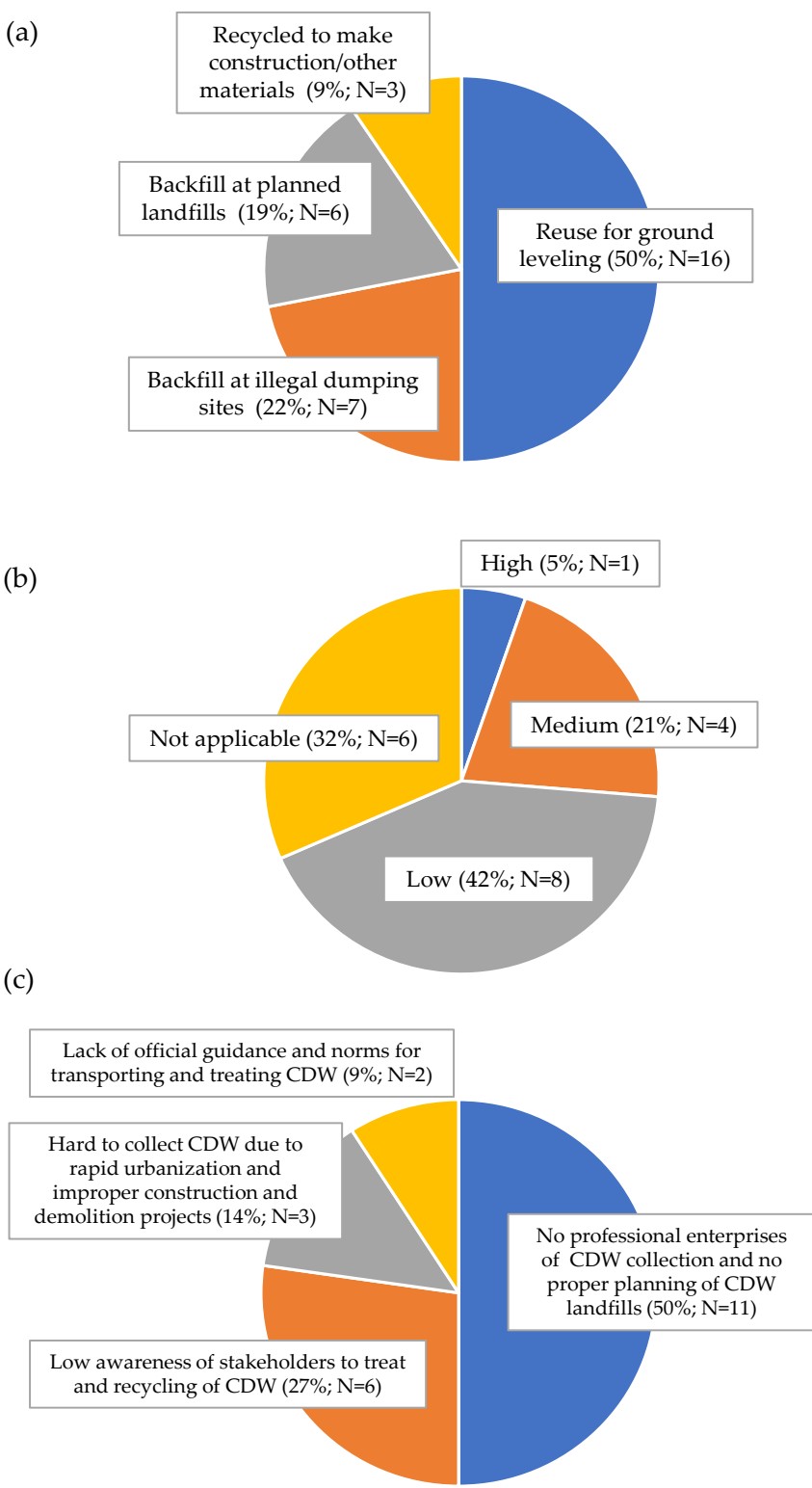

**Figure 2.** Results from interview survey of local authorities. (**a**) Methods of treatment and recycling of CDW (selected multiple answers), (**b**) level of compliance/implementation to the regulation on construction solid waste management (Circular No. 08/2017/TT-BXD [27]), and (**c**) difficulties and obstacles in the management and recycling of CDW (selected multiple answers). N: number of answers.

**Table 3.** Results from interview survey of construction enterprises (n = 10). Linguistic variables used to calculate numerical values are also given.

| S/N | Item | Numerical Value | Translation into Opinion |
|---|---|---|---|
| Option to current legislation/legal framework on CDW | | | |
| 1 | Ease of following | 0.13 | Totally disagree |
| 2 | Appropriation | 0.35 | Either neutral or disagree |
| 3 | Economic efficiency | 0.15 | Totally disagree |
| 4 | Necessity | 0.73 | Agree |
| Main difficulties to sort demolished waste | | | |
| 1 | Cost increase | 0.93 | Totally agree |
| 2 | Manpower shortage | 0.43 | Neutral |
| 3 | Time consumption | 0.80 | Agree |
| 4 | Lack of skill and technology | 0.15 | Either disagree or totally disagree |
| 5 | Lack of machinery | 0.15 | Either disagree or totally disagree |
| 6 | Limited site area | 0.05 | Totally disagree |
| 7 | Lack of legal guidelines | 0.23 | Disagree |
| 8 | Lack of post-demolition treatment facility | 0.55 | Neutral |
| 9 | Lack of awareness of workers | 0.00 | Totally disagree |

Linguistic variables

| 0.0 | 0.25 | 0.5 | 0.75 | 1.0 |
|---|---|---|---|---|
| Totally disagree | Disagree | Neutral | Agree | Totally agree |

For promoting the recycling activities of CDW, a set of four options were suggested to interviewees regarding possible actions. The results show that the majority of interviewees admitted ignorance about recycling CDW, with 16 of 19 or 84% of interviewees saying that there are no tools to guide the development of CDW recycling, prioritized policies, supporting, or consulting for local recycling enterprises. Only two respondents in two districts (10.5%) agreed that supporting policies are necessary in terms of investment, loans, and prioritized land rental. Three respondents in three districts thought that education is necessary to raise the awareness of people and enterprises about sorting, collecting, and recycling CDW. Only one respondent considered it necessary to enhance advertising, price subsidy, product sales, or tax incentives to recycle products. Two districts chose all three action options, but they simultaneously declared that such policies would only be feasible in about 5–10 years. Two rural districts had the same reason for selecting "Nothing" as their current CDW is fully reused for leveling, so there is no need for CDW recycling in the foreseeable future, and sometimes new constructions even have to search for CDW to buy for leveling. It is sold at roughly 80,000–100,000 Vietnam Dong (3.5–4.5 US dollars) per truckload for a volume of 5–10 m$^3$.

*3.3. Current Condition of CDW Landfill/Dumping Sites*

A basic information survey was carried out to collect fundamental data on CDW dumping sites in Hai Phong. The results show that the city has a planned landfill operation at Gia Minh, Thuy Nguyen District. This complex is operated by Haiphong Urban Environment One Member Limited Company (Haiphong, Vietnam) to treat all municipal solid waste, not only CDW. Also, in this district, EIB mentioned another CDW landfill operated by a private company (Phu Hung Jsc., Haiphong, Vietnam) with an area of two hectares. Hai Phong used to have an official CDW landfill at Trang Cat in the Hai An district, but it had been already developed into a new project at the time of this study. Cat Hai district reported two solid waste landfills on the island. Of these, the general landfill, which is roughly three hectares, accepts both MSW and CDW. The other one is at the Tan Chau commune with an area of about five hectares. This is planned to be finished by 2022, but it is not yet operational. Illegal dumping is reported in 6/15 surveyed districts (a rate of 40%), mostly from urban districts (5/7, 71%). Interviewees remarked that "illegal dumping

sites are scattered, the amount per site is also not large—usually several piles of CDW". Thus, it is complicated to manage and penalize dumping. Meanwhile, the officers of Bach Long Vi district reported that CDW generation on the island is very limited because there are no major construction/demolition works. It is due to the small number of buildings, and all CDW generated is employed for backfilling. Therefore, there is no illegal dumping on this island.

In this study, a total of nine landfill locations were visited using the check sheet survey to clarify the current situation of CDW dumping sites in Hai Phong (Table 4). The earth view and typical photos of official waste treatments are shown in Figure 3, and illegal dumping sites are shown in Figure 4. Through the site visits to all notable landfills, it is easy to observe that there is no official landfill dedicated to CDW. All operating landfills/treatment plants (namely Minh Tan, Dinh Vu, and Trang Cat) mainly focus on municipal solid waste, whereas it appears the planned treatment complex (Gia Minh) will focus on treating MSW and wastewater. All current official landfills have large areas, ranging from more than 10 hectares to almost 40 ha. The Trang Cat treatment complex was built with aid from Korea, while the Gia Minh treatment complex is being built with aid from Japan. It can be concluded that Hai Phong PC has been active in finding sustainable solutions for solid waste management. However, CDW has not received much attention so far. The collection and transportation of CDW are concentrated at the planned dumping sites without mandatory sanctions. Therefore, the CDW at the dumps is almost nonexistent. Illegal dumping is still the primary method of CDW disposal, and this method is still preferred due to its capability for reclaiming land (dumping into the river) and backfill for local people when they want to build houses on previous paddy fields. Furthermore, due to the rapid urbanization of a developing city, CDW may soon become an environmental challenge for Hai Phong City. Therefore, the management agencies of this city need to issue sanctions and use inspection measures in CDW management more effectively.

**Table 4.** Summary of waste treatment plants that accept CDW and illegal dumping sites of CDW in Hai Phong city.

| | Name/ District | Gia Minh Waste Treatment Complex/Thuy Nguyen | Minh Tan Waste Treatment Plant/Thuy Nguyen | Dinh Vu Waste Management and Treatment Plant/Hai An | Trang Cat Solid Waste Treatment Complex/ Hai An | Xuan Dam/ Cat Hai | Luu Kiem/ Thuy Nguyen | May Chai/ Ngo Quyen | Nam Hai/ Hai An | Le loi/ An Duong |
|---|---|---|---|---|---|---|---|---|---|---|
| 1 | Operation years | Not yet | 2 | 5 | 7 | 3 | Unknown | Unknown | Unknown | Unknown |
| 2 | Landowner | Haiphong PC | Haiphong PC | Haiphong PC | Haiphong PC | District PC | Illegal dumping | Illegal dumping | Illegal dumping | Illegal dumping |
| 3 | Estimated land area | 37.4 ha | 12.7 ha | 32,500 m$^2$ | 31.6 ha | ~3 ha | 1340 m$^2$ | 360 m$^2$ | small | 2380 m$^2$ |
| 4 | Previous land use | swamp area and garbage dump | swamp area and garbage dump | Unknown | Swamp land | Vacant land | Paddy field | River bank | Road side | Road side |
| 5 | Dumped waste | MSW, CDW (in the future) | MSW, industrial waste (including hazardous waste) | MSW, insignificant amount of CDW | MSW and CDW | MSW and CDW | CDW | CDW | CDW, domestic waste (in small amount by surrounding residence) | CDW |
| 6 | Daily intake | Unknown | 100–150 tons MSW and industrial waste/day | 200 tons of MSW/day | 400–500 tons of MSW mixed with limited CDW/day | Unknown | Unknown | Unknown | Unknown | Unknown |
| 7 | Height and/or depth of dumped waste | 0 | <1 m | 2–3 m | 10–15 m | 1–2 m | <1 m | <1 m | <1 m | 1–1.5 m |

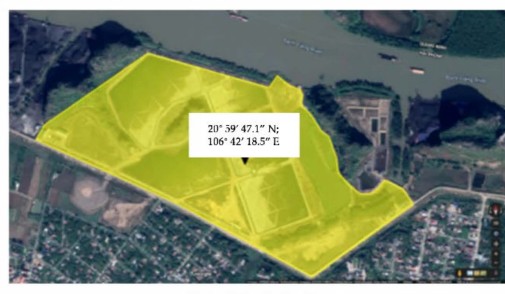 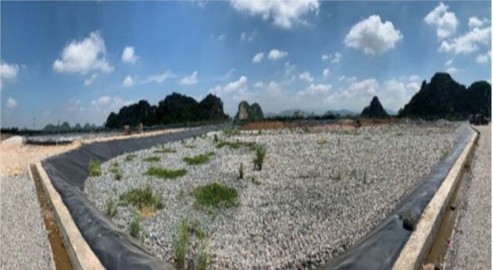

(a) Gia Minh waste treatment complex, Thuy Nguyen district

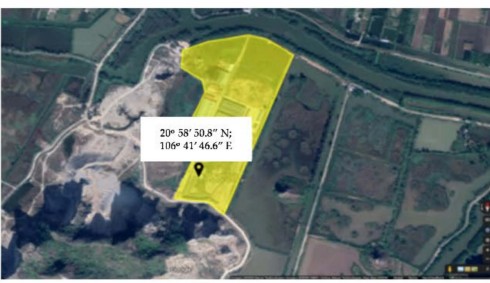 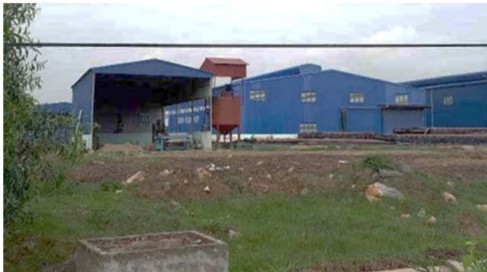

(b) Minh Tan waste treatment plant, Thuy Nguyen district

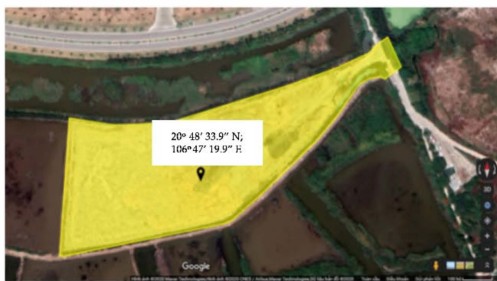 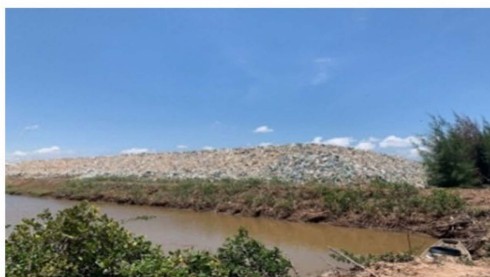

(c) Dinh Vu waste management and treatment plant, Hai An district

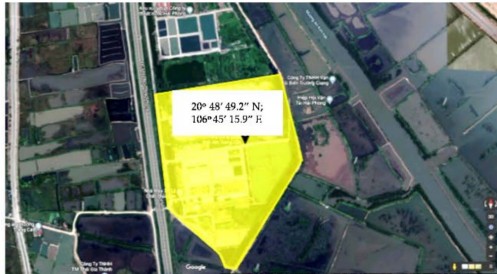 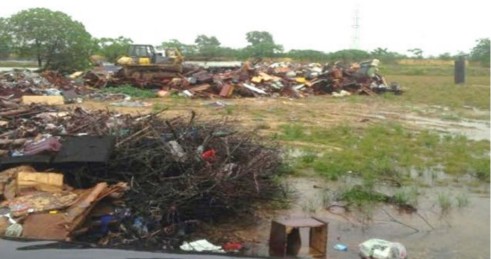

(d) Trang cat solid waste treatment complex, Hai An district

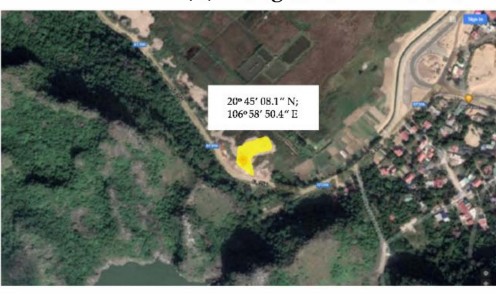 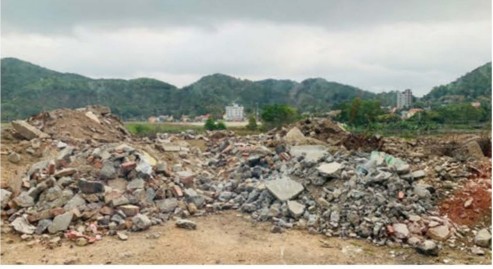

(e) Cat Hai district

**Figure 3.** Earth view and photo of waste treatment plant/complexes that accept CDW in Hai Phong City (map datas © 2021 Google).

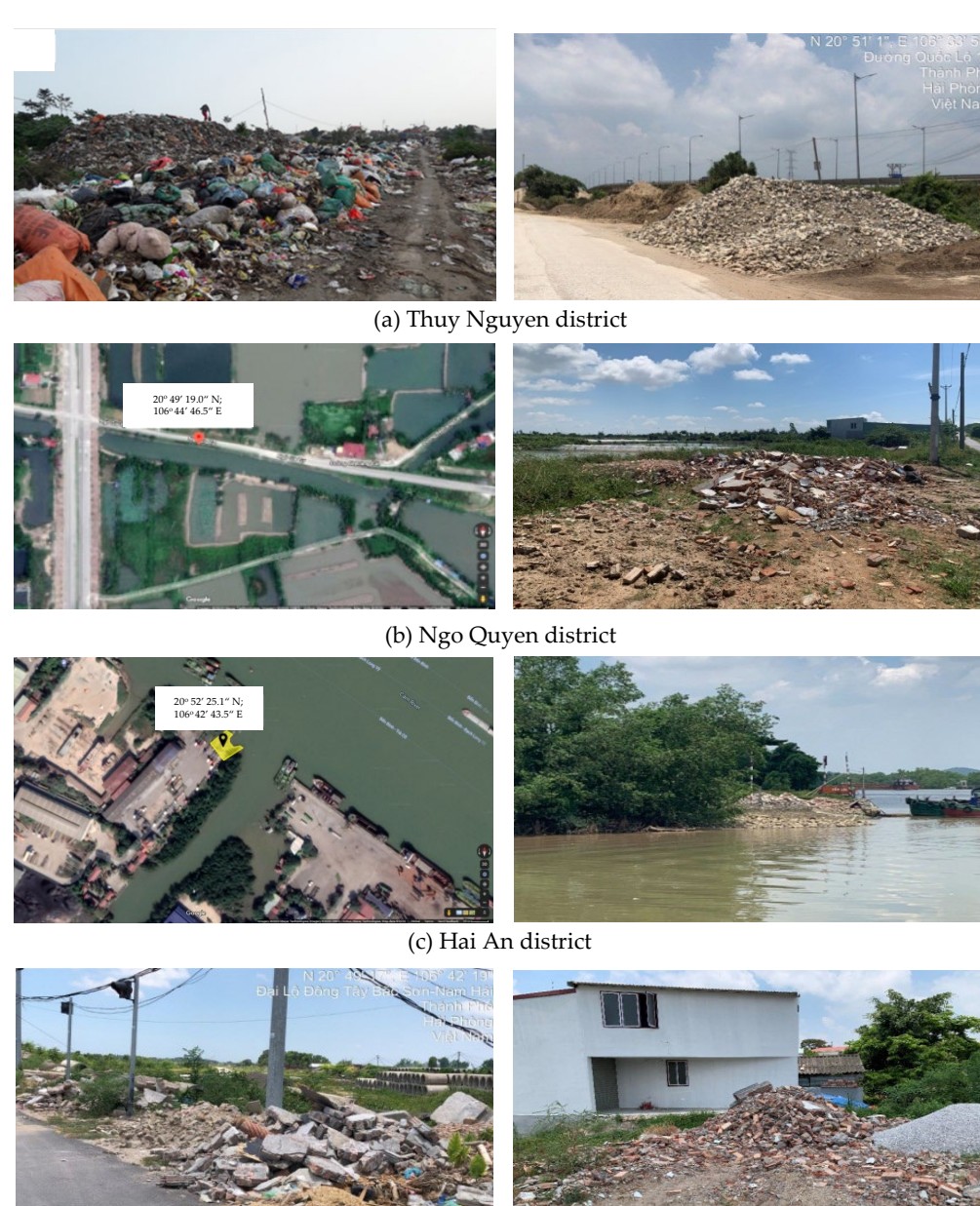

**Figure 4.** Earth view and photo of illegal CDW dumping sites in Hai Phong City (map data © 2021 Google).

*3.4. Construction and Demolition Waste Generation Rates*

3.4.1. WGR of CDW from Old Buildings to Be Demolished

According to Decree 101/2015/NĐ-CP [39], Hai Phong City has 205 large old buildings, of which 178 will be demolished to build new apartments in the coming years. A direct survey of 73 large old buildings of varying ages due for demolition was made in five urban districts and three rural districts, as indicated in Figure 1. The majority of them are old apartment buildings (67/73 or 92%), and the rest are office buildings and schools. The existing structures are mostly cast-in-situ or precast concrete and clay brick partition walls. The surveyed buildings typically have three to four floors, and the total floor area and average GFA of the old large surveyed buildings by construction decades 50s, 60s, 70s, 80s, and 90s are shown in Figure 5. It is clear that the number of apartment buildings built in the 60s and 70s makes up a significant portion of surveyed buildings (55/73 buildings or

75.34%). Until now, after 50–70 years since being commissioned, most buildings are heavily and dangerously degraded (concrete cover is spalling, reinforcing steel is corroded), thus in dire need of being demolished and rebuilt as soon as possible. This fact will lead to the generation of large old building demolition waste in the coming years.

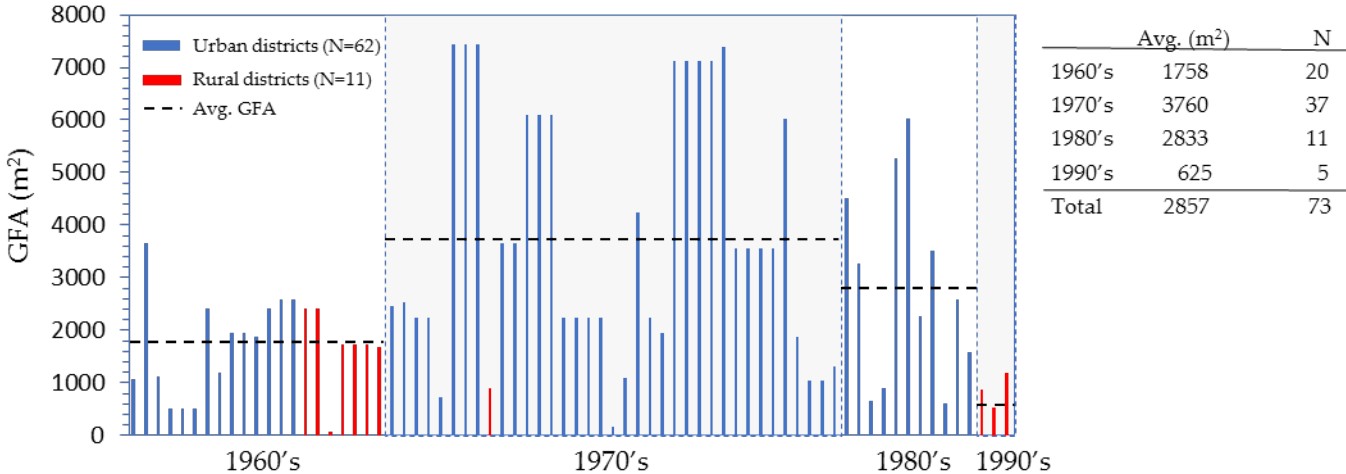

**Figure 5.** Gross floor area (GFA) of large old buildings surveyed in this study (1960's–1990's). N is number of samples.

Figure 6a shows an old building that was surveyed to verify the WGR from demolition. This was a four-story reinforced concrete (RC) building with $16.1 \times 13.7$ m$^2$ of brick walls, and a GFA of 882.4 m$^2$. It was inaugurated and had been operating since around 1990. Hence, at the moment of demolition, this building was approximately 30 years old. Based on the as-built drawings of this building and on-site measurements, the amount of CDW during demolition was quantified and calculated as summarized in Table 5. This study focused only on the main components such as CC, CB, and steel at the demolition site. Other components (toilet, washbasin, … ) that were small in volume or reused were omitted.

Table 5 shows that the generation coefficients CC, CB, and steel of the works were 0.422, 0.627, and 0.00278 tons/m$^2$, respectively. Thus, the generation rate of CDW from the demolition of the surveyed old building is $G^{OLB}_{Demolition} = 1.053$ tons/m$^2$. This result is much larger than that obtained from the study conducted in Hanoi by Hoang et al. [10]. It is worth mentioning that CB is 1.5 times heavier than CC and the amount of reinforcement used was very small, about 3 kg/m$^2$, consistent with the building structure in Hai Phong. This is because the load-bearing wall structure area in old apartments and offices is very large, which is also typical for old public buildings in this locality.

In this study, the $G^{OSB}_{Demolition}$ was estimated to be constant relative to $G^{NSB}_{Demolition}$, although, in fact, $G^{OSB}_{Demolition}$ may be larger because the small old houses (built before 2010) are often designed with load-bearing walls and during use, people often renovate/repair, leading to an increase in the wall waste composition compared to the $G^{NSB}_{Demolition}$, which is calculated directly from construction drawings. However, residential/small buildings built after 2010 often have reinforced concrete frame structures so the volume of reinforced concrete is higher and the masonry wall acts only as a partition and cover structure with a low volume.

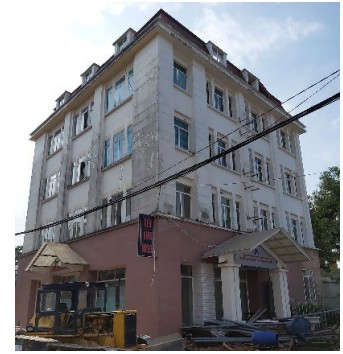
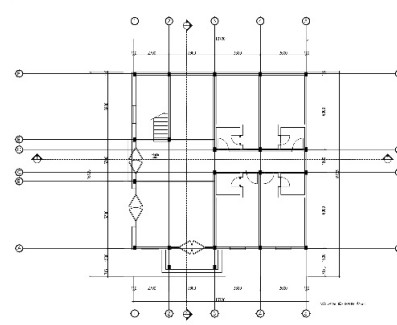

(**a**) Photo and as-built drawing (1st floor) of a demolished building (in December 2020)

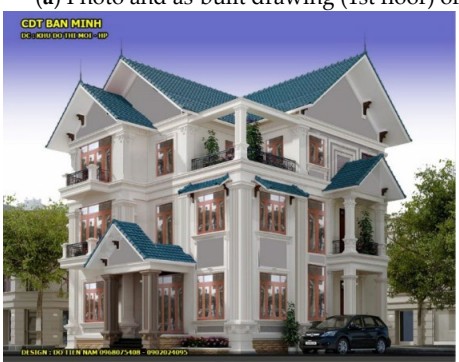
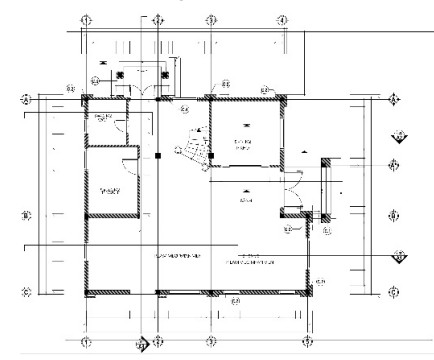

(**b**) Perspective view and as-built drawing (1st floor) of newly built villa

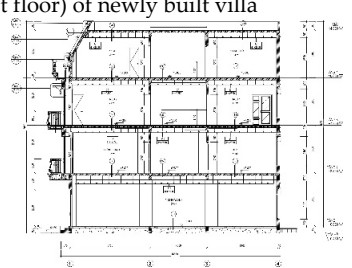

(**c**) As-built drawings (left: 1st floor, right: elevated section) of newly built shophouse

**Figure 6.** (**a**) Demolished building and (**b**,**c**) newly constructed buildings used to calculate the waste generation rate (WGR in tons/m$^2$) in this study (in December 2020).

### 3.4.2. WGR from New Construction

The CDW from new construction includes CDW produced during construction and the amount of CDW that will be generated when it is demolished in the future. To determine these CDW amounts, two individual houses that were licensed for construction in the last three years were investigated. The first project was a villa house in the New Urban Area. It was constructed in 2017 as a three-story RC building with a plan area of $12.4 \times 10.8$ m$^2$ and a measured GFA of 429 m$^2$ (Figure 6b). The second project was a four-story RC shophouse constructed in 2019 in the Trang Due Urban Area (Figure 6c). This house has an area of $15.7 \times 6.0$ m$^2$ with a measured GFA of 376.8 m$^2$. Both are private houses and classified as small buildings with the GFA below 500 m$^2$. A quantity survey was performed on the set of as-built drawings provided by the structural designer and the contractor's answers. The detailed calculation data are summarized in Table 5. The average WGR during the construction of these new small buildings is $G^{NSB}_{Construction} = 34.5 \times 10^{-3}$ tons/m$^2$, and the future WGR from the demolition by the end of their service life (assuming 30 years) is $G^{NSB}_{Demolition} = 0.758$ tons/m$^2$. A comparison of WGRs from construction and demolition work is shown in Table 6. It is clear that construction WGR is much lower than demolition WGR, and the WGR values in the studies also vary widely depending on the type and function of the building or building structures [10,16,44]; amount of waste generated from construction, demolition, or renovation activities [45,46]; construction methods [47]; and CDW components by region/country [1,29,48].

**Table 5.** Waste generation rate (WGR) from a demolished old building and newly built buildings (villa and shophouse) in this study.

| Item | CC (m3) | CB (m3) | Steel (tons) | Total (CC+CB+S) |
|---|---|---|---|---|
| Demolished old building | | | | |
| Demolished waste | 169.3 | 368.4 | 2.45 | |
| WGR (tons/m$^2$) | $4.22 \times 10^{-1}$ | $6.27 \times 10^{-1}$ | $2.78 \times 10^{-3}$ | 1.053 |
| Newly built buildings | | | | |
| (1) Villa | | | | |
| During Construction | 4.7 | 3.6 | 0.060 | |
| WGR (tons/m$^2$) | $2.41 \times 10^{-2}$ | $1.26 \times 10^{-2}$ | $1.40 \times 10^{-4}$ | 0.037 |
| When demolished | 92.0 | 80.8 | 16.22 | |
| WGR (tons/m$^2$) | $4.72 \times 10^{-1}$ | $2.83 \times 10^{-1}$ | $3.78 \times 10^{-2}$ | 0.793 |
| (2) Shophouse | | | | |
| During Construction | 3.8 | 2.5 | 0.046 | |
| WGR (tons/m$^2$) | $2.20 \times 10^{-2}$ | $1.00 \times 10^{-3}$ | $1.20 \times 10^{-4}$ | 0.032 |
| When demolished | 74.3 | 64.4 | 12.63 | |
| WGR (tons/m$^2$) | $4.34 \times 10^{-1}$ | $2.56 \times 10^{-1}$ | $3.35 \times 10^{-2}$ | 0.723 |

It is clear that the $G_{Demolition}^{OSB}$ and $G_{Demolition}^{NSB}$ results are very different from the values of WGR at small and large demolition sites of 610 and 318 kg/m$^2$ or 1192 and 510 kg/m$^2$ obtained from the studies conducted by Hoang et al. [10] or Nghiem et al. [16] for Hanoi City. And $G_{Construction}^{NB}$ at small construction has a much bigger but much smaller than the result of N.H. Hoang et al. [10] seveyed in small and large construction sites. This is because only three material components, concrete, brick mortar, and steel waste, were included in the calculation of emission factors during new construction in Hai Phong City, but not the amount of soil waste as calculated by Hoang et al. [10]. In addition, the villas and shophouses in this study are privately owned, so the better construction quality leads to a lower amount of waste generated during construction.

**Table 6.** Comparison of waste generation rate (WGR) in this study to values reported in the literature.

| Region/Country | WGR (tons/m$^2$) | | Ref. |
|---|---|---|---|
| | Construction Waste from Newly Built Building | Demolished Waste from Old Building | |
| Hong Kong/China | 0.201–0.297 [2] (reinforced concrete) | 0.565–0.918 [2] (reinforced concrete) | [12,13] |
| Oslo, Trondheim/Norway | 0.0294 [1] (reinforced concrete, wood) | 0.575 [1], 1.103 [2] (reinforced concrete, wood) | [14] |
| Shenzhen/China | 0.0033–0.0088 [2] (reinforced concrete, timber/plywood) | - | [44] |
| EU | 0.018 [1], 0.033–0.040 [2] (reinforced concrete) | 0.401–0.492 [1], 0.768–0.840 [2] (reinforced concrete) | [45] |
| Many states/USA | 0.021 [1], 0.025 [2] | 0.024 [1,3], 0.384 [2,3] | [46] |
| Malaysia | 0.099–0.033 [2] (reinforced concrete, timber/plywood) | 1.04 [2] (reinforced concrete) | [47] |
| Kyoto, Osaka, Hyogo/Japan | - | 0.049 (wooden), 0.810 (steel), 1.580 (reinforced concrete), 2.02 (steel reinforced concrete) | [48] |
| Japan | 0.030 (wooden), 0.016 (non-wooden) | 0.390 (wooden), 0.880 (non-wooden) | [21] |
| Hanoi/Vietnam | 0.0079 [1], 1.030 [2] (reinforced concrete) | 0.061 [1], 0.032 [2] (reinforced concrete) | [10] |
| Hanoi/Vietnam | - | 1.192 [1], 0.510 [2] (reinforced concrete) | [16] |
| Hai Phong/Vietnam | 0.032–0.037 [1] (reinforced concrete) | 0.723–0.793 [1], 1.053 [2] (reinforced concrete) | This study |

The volumetric density of CDW is taken as 1.413 tons/m$^3$. [1] Small-scale construction and demolition work: typically, private houses with land area < 500 m$^2$. [2] Large-scale construction and demolition work: typically, public houses, new urban areas, and commercial center. [3] No information on building structure.

*3.5. CDW Generation Projections*

3.5.1. Future Waste Projection from New Construction Work

The results of multiple regression analysis according to the variables are reported in Table 7. From Table 7, we see that the multiple regression Equation (12) has the multiple R = 0.950; $R^2$ = 0.902; standard error = 165. This means that the equation with two variables $X_1$ is the P rate, and $X_2$ is the GRDP, bringing useful results. In addition, when testing the results of the regression analysis of this combination against the F-test standard, the predicted F value is 36,649, much larger than the critical value of 6.013 for 1% risk. The results of the comparison between the predicted total floor area value and the actual value are shown in Figure 7. The results show that the predicted value is quite close to the actual value, and the average relative error is only 0.69%. This result is consistent with the results of MLIT [21]. Therefore, the regression given in Equation (12) is sufficiently reliable:

$$[\text{GFA estimation}] = 2.74 \times 10^2 \times [\text{P rate}] + 7.27 \times 10^{-3} \times [\text{GRDP}] + 428 \qquad (12)$$

From Equation (12), we see that the coefficients of the variable P rate of $2.74 \times 102$ are about 37,802 times higher than the coefficients of the GRDP variable (only $7.27 \times 10^{-3}$). However, in terms of GRDP value, it is 267,091 times larger than the P rate value (only approximating 1.0%, for example, in 2020). According to the statistical results from 1999 to 2020, the average P rate is 0.97%/year; with the maximum possible value of 1.42% and the minimum value of 0.15%, the GFA value is predicted to increase or decrease by 8.9% and 16.3%, respectively, compared to the GFA at P rate and average GRDP. When GRDP is constant, then GFA increases by 24.5% if the P rate increases by 1.0%. In contrast, the average GRDP value in the years from 1999 to 2020 reached 95,085.51 billion Dong (equivalent to the average GRDP growth of 12.57%/year). The highest rate of GRDP increase is 23.20% and the smallest is 7.05%, so the GFA is forecasted to increase by 90.2% and decrease by 38.2% compared to GFA at GRDP and the average P rate. When the P rate is constant, if GRDP increases by 10%, then GFA increases by 5.0%. Therefore, the GFA is mainly dependent on the P rate change rather than the GRDP.

According to the report on adjustment of general construction planning of Hai Phong City to 2040, vision to 2050 [49], the P rate in 2019 is 0.8%/year and the GRDP in 2018 was $7.9 billion (the capita income was $4277/person). The indicators of population planning and economic development follow the three scenarios shown in Table 8. Using the proposed CDW forecasting model, the total constructed floor area forecast for 2050 is shown in Figure 8. Using the WGR during the construction process is $G_{Construction}^{NSB} = 34.5 \times 10^{-3}$ tons/m$^2$, the amount of construction waste generated in the short (5–10 years) and long (20–30 years) term are shown in Figure 9. It is noted that this WGR will be lower than that of large projects during construction due to strict quality control during the construction of private houses.

**Table 7.** Statistical parameters and coefficients from multiple regression analysis in Equation (11).

| | $X_1$ and $X_2$ | | | |
|---|---|---|---|---|
| | GRDP and P | P and GRDP Rate | GRDP Rate and P Rate | P rate and GRDP |
| | Statistical parameters | | | |
| Multiple R | 0.969 | 0.933 | 0.744 | 0.950 |
| $R^2$ | 0.939 | 0.870 | 0.554 | 0.902 |
| Adjusted $R^2$ | 0.924 | 0.838 | 0.442 | 0.877 |
| Standard Error | 130 | 189 | 351 | 165 |
| Observations | 11 | 11 | 11 | 11 |
| | Coefficients | | | |
| $a_1$ | $1.16 \times 10^{-2}$ | 6.93 | $-67.6$ | $2.74 \times 10^2$ |
| $a_2$ | $-5.50$ | $-4.98 \times 10$ | $-1.66 \times 10^3$ | $7.27 \times 10^{-3}$ |
| b | $1.09 \times 10^4$ | $-1.12 \times 10^{-4}$ | $4.34 \times 10^3$ | $4.28 \times 10^2$ |

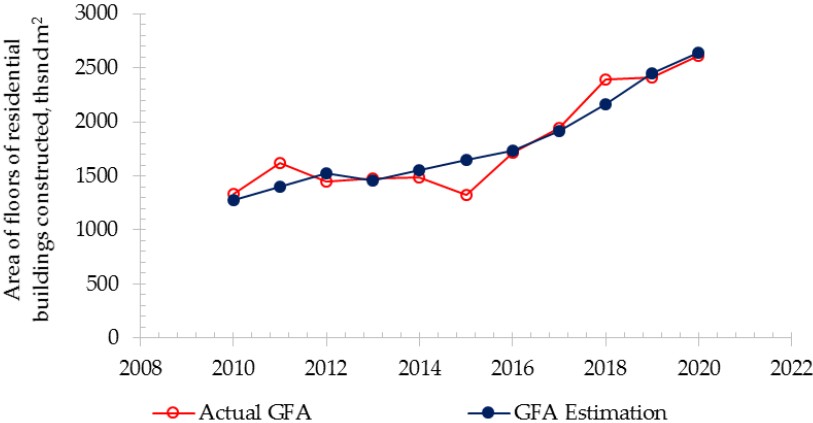

**Figure 7.** Comparison of actual and predicted areas of the total building floors.

The results show that CDW output in the city will continue to increase rapidly in the next 5 and 10 years. It will reach 139–160 thousand tons by 2025 and 178–219 thousand tons by 2030, corresponding to an increase of 49–70 thousand tons and an increase 2.0–2.4 times higher than that in 2020. This is consistent with the pace of urbanization and the city's plan to build key works in the coming years. Hai Phong City needs not only to provide regulations, solutions to minimize CDW release during construction, but also to accelerate the application of CDW recycling treatment methods and promote the use of CDW waste sources as construction materials to ensure sustainable economic development.

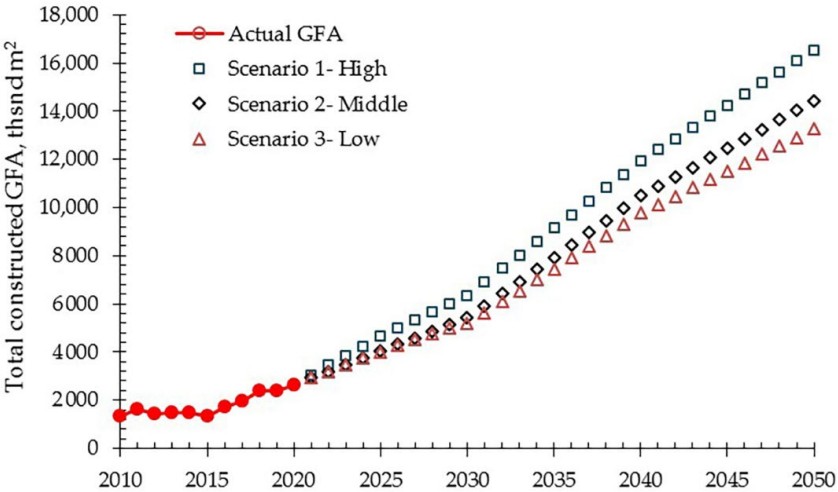

**Figure 8.** Projected total constructed GFA in Hai Phong City by 2050.

**Table 8.** Population and economic development norms according to planning scenarios.

| Planning Criteria | Scenario 1: High Level | Scenario 2: Middle Level | Scenario 3: Low Level |
|---|---|---|---|
| Population growth, %/year | | | |
| 2019–2025 | 2.3 | 1.6 | 0.9 |
| 2025–2030 | 2.5 | 1.7 | 0.9 |
| 2030–2035 | 2.9 | 2.0 | 1.1 |
| 2035–2040 | 3.3 | 2.4 | 1.5 |
| 2040–2050 | 3.5 | 2.9 | 2.3 |
| Economic development forecast according to the master plan [47], USD/person | | | |
| 2020–2030 | 16,214 | 14,740 | 13,266 |
| 2030–2040 | 32,890 | 29,900 | 26,910 |
| 2040–2050 | 47,000 | 41,445 | 3,7300 |

3.5.2. Future Waste Projection from Demolition of Old Buildings

Based on the obtained WGRs, we can predict the amount of CDW generated by the demolition of old structures in the coming 5–30 years according to the following formula:

$$M_{Demolition}^{OB} = G_{Demolition}^{OLB} \times GFA_{ave}^{OLB} \times N + G_{Demolition}^{OSB} \times GFA^{OSB} \qquad (13)$$

where $N$ is the number of old buildings expected to be demolished in the next 5–30 years.

According to the National Technical Regulation No. 03/2012/BXD [50], grade II works have a service life of 50 to 100 years. Assuming the service life of a work is 60 years, works built in the 1950s and 1960s are coming to the end of their service life, so they will be demolished in the next 5 years (by 2025), and the works built in the 1970s, 1980s, and 1990s will be demolished in next 10, 20, and 30 years (by 2030, 2040, 2050), respectively [50]. The total amount of demolition waste from large old buildings in the next 30 years will be 593,607 tons, an average of about 19,787 tons/year.

The small old buildings built before 1990 are mainly residential/private houses and belong to the grade IV works with a lifespan of less than 20 years [50], so most of these houses have been demolished or were destroyed by the war. Therefore, the amount of waste generated by these works will be insignificant from 2020. From 1990 to 2020, the economy of Vietnam and Hai Phong City was better, and the buildings were constructed more solidly and may reach up to level III durability with a lifespan of 20 to 50 years according to the National Technical Regulation No. 03/2012/BXD [50]. Assuming their lifespan is 30 years, these houses will be demolished in the years from 2020 to 2050. There is no statistical data on housing before 2010, so that the GFA of residential houses built from 1990–2010 will be predicted according to three scenarios with rates of 70% (scenario 1—high rate), 50% (scenario 2—middle) and 30% (scenario 3—low) of the total GFA construction. The total GFA built in the years 1990–2010 will be forecasted by the multivariable regression of Equation (12) with the values of P rate and GRDP in those years (Table 9).

**Table 9.** GRDP, P rate data, and GFA estimation of old buildings from 1990–2009.

| Year | GRDP, (Bil. Dongs) | P Rate, (%) | Total GFA Estimation, (Thsnd. m²) | GFA Estimation of Old Small Buildings, (Thsnd. m²) | | |
|------|------|------|------|------|------|------|
| | | | | Scenario 1 | Scenario 2 | Scenario 3 |
| 1990 | 8467.9 | 2.43 | 1157.7 | 810.403 | 578.859 | 347.316 |
| 1995 | 13,331.9 | 2.36 | 1173.8 | 821.687 | 586.919 | 352.151 |
| 1999 | 22,219.8 | 0.98 | 859.2 | 601.469 | 429.621 | 257.772 |
| 2000 | 24,441.8 | 1.06 | 896.2 | 627.325 | 448.089 | 268.853 |
| 2001 | 26,978.8 | 0.99 | 895.4 | 626.760 | 447.686 | 268.612 |
| 2002 | 29,852.1 | 1.10 | 947.3 | 663.136 | 473.668 | 284.201 |
| 2003 | 33,044.0 | 1.40 | 1053.0 | 737.122 | 526.516 | 315.910 |
| 2004 | 36,840.4 | 0.95 | 957.8 | 670.484 | 478.917 | 287.350 |
| 2005 | 41,259.9 | 0.15 | 768.7 | 538.087 | 384.348 | 230.609 |
| 2006 | 46,434.2 | 0.87 | 1006.1 | 704.290 | 503.064 | 301.838 |
| 2007 | 52,348.1 | 0.98 | 1079.3 | 755.503 | 539.645 | 323.787 |
| 2008 | 59,102.3 | 0.97 | 1125.8 | 788.026 | 562.875 | 337.725 |
| 2009 | 63,584.1 | 0.90 | 1136.8 | 795.788 | 568.420 | 341.052 |

Forecasts of the total amount of construction waste generated in Hai Phong City by 2050 are shown in Table 10 and Figure 9. Obviously, the rate of CDW generation increases on average 3–7%/year in the period 2020–2040, but increases sharply by 20.5%/year in the period 2040–2050. This is because the speed of urbanization and development of the construction industry in Hai Phong increased very rapidly in the years 2015–2020. The total CDW amount expected to be generated in 2050 in Hai Phong will reach 2,510,575 tons (6878 tons/day). This is a huge amount that will need to be treated in the future. It is worth noting that the WGR used in this calculation is only 0.0345 tons/m² during construction and 0.758 tons/m² during demolition. If we use the public works WGR when demolishing

(1.053 tons/m$^2$) and include CDW when repairing existing construction, then the total amount of CDW generated is much larger. Therefore, it is urgent that Hai Phong City develop planning for storage yards, as well as foster technologies to effectively treat and recycle this huge amount of CDW in the coming years.

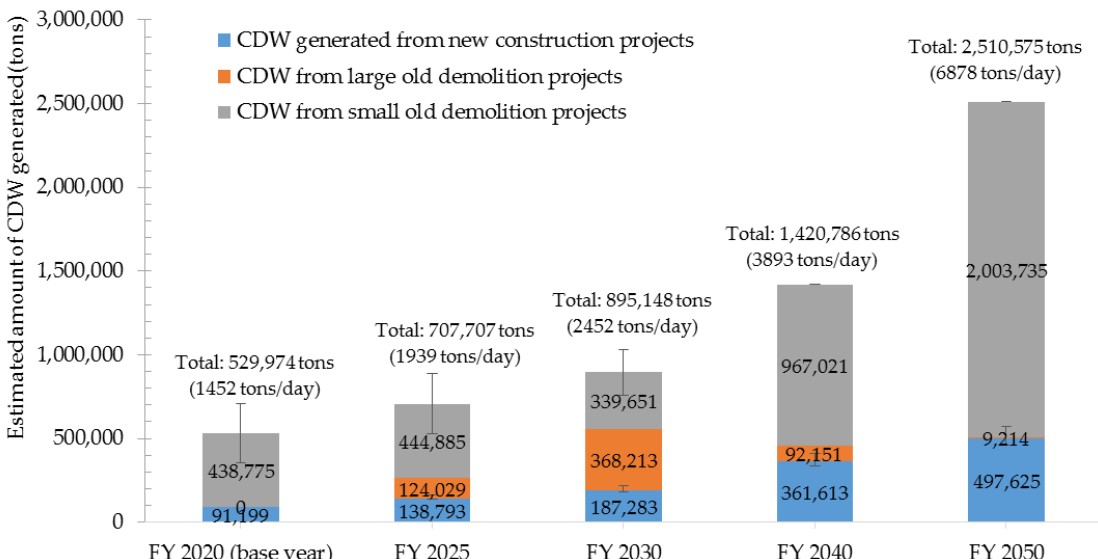

**Figure 9.** Estimated total amount of CDW generated in Hai Phong City by 2050.

**Table 10.** Predicted and projected results of CDW generated during construction and demolition from 1990–2050.

| Year | | GFA, Thsnd. m$^2$ | CDW Generated during New Construction, tons | CDW Generation from Demolition of Old Large Buildings, tons | CDW Generation from Demolition of Old Small Buildings, tons |
|---|---|---|---|---|---|
| FY1990 prediction | Scenario 1—High | 810.403 | 27,959 | - | 614,285 |
| | Scenario 2—Middle | 578.859 | 19,971 | - | 438,775 |
| | Scenario 3—Low | 347.316 | 11,982 | - | 263,265 |
| FY 1995 prediction | Scenario 1—High | 821.687 | 28,348 | - | 622,839 |
| | Scenario 2—Middle | 586.919 | 20,249 | - | 444,885 |
| | Scenario 3—Low | 352.151 | 12,149 | - | 266,931 |
| FY 2000 prediction | Scenario 1—High | 627.325 | 21,643 | - | 475,512 |
| | Scenario 2—Middle | 448.089 | 15,459 | - | 339,651 |
| | Scenario 3—Low | 268.853 | 9275 | - | 203,791 |
| FY 2010 actual results | | 1275.754 | 44,014 | - | 967,021 |
| FY 2015 actual results | | 1647.213 | 56,829 | - | 1,248,588 |
| FY 2020 actual results | | 2643.450 | 91,199 | - | 2,003,735 |
| FY 2025 projection | Scenario 1—High | 4645.161 | 160,258 | 124,029 | 3,521,032 |
| | Scenario 2—Middle | 4022.988 | 138,793 | 124,029 | 3,049,425 |
| | Scenario 3—Low | 4017.036 | 138,588 | 124,029 | 3,044,913 |
| FY 2030 projection | Scenario 1—High | 6343.427 | 218,848 | 368,213 | 4,808,318 |
| | Scenario 2—Middle | 5428.499 | 187,283 | 368,213 | 4,114,802 |
| | Scenario 3—Low | 5173.051 | 178,470 | 368,213 | 3,921,172 |
| FY 2040 projection | Scenario 1—High | 11,940.245 | 411,938 | 92,151 | 9,050,705 |
| | Scenario 2—Middle | 10,481.543 | 361,613 | 92,151 | 7,945,009 |
| | Scenario 3—Low | 9764.755 | 336,884 | 92,151 | 7,401,684 |
| FY 2050 projection | Scenario 1—High | 16,544.814 | 570,796 | 9214 | 12,540,969 |
| | Scenario 2—Middle | 14,423.926 | 497,625 | 9214 | 10,933,336 |
| | Scenario 3—Low | 13,252.287 | 457,204 | 9214 | 10,045,234 |

## 4. Conclusions

Based on the results obtained, the conclusions and lessons learned are as follows:

Limited attention is currently paid to CDW management in Hai Phong City and in Vietnam. As a result, there is no specialized management agency, and waste generation statistics are insufficient. The information and compliance with policies on CDW management are at a low level. The CDW management and recycling in localities still face many difficulties, including: (i) there is no professional enterprise gathering CDW and no proper planning for CDW landfilling; (ii) awareness of stakeholders of the importance of treating and recycling CDW is low; and (iii) official guidance in CDW transportation and treating, and the official norms thereof, is lacking. Overcoming these difficulties will promote the establishment of an effective and sustainable management system for CDW.

As-built drawing analysis in combination with site visitation is an appropriate method to estimate CDW generation. The generation rate of CDW when demolishing public old houses is 1.053 tons/m$^2$, of which CB accounts for the highest share (0.63 tons/m$^2$). The CDW generation rate from newly built urban housing is only about 0.758 ton/m$^2$ when demolishing or 0.0345 ton/m$^2$ when constructing. Demolished materials are mainly reused for site leveling. Illegal dumping is quite common due to the lack of centralized waste disposal sites. This raises a need for the development of CDW recycling technologies and facilities, technical standards for recycled CDW products, and a formal market thereof.

In this study, a predictive model of total building floor area depending on GRDP and P rate by the multivariable regression analysis method was proposed. In order to predict the CDW generation by this model, the case study in Hai Phong City was selected to empirically identify the WGRs from new constructions and old demolition projects. This proposed model has an advantage in accurately forecasting the total amount of CDW over time based on the statistical data on economic development and annual population growth of localities in the past, present, and future. Finally, the CDW estimation method provided in this study is expected to fill a data gap in Vietnam, including Hai Phong City, and would contribute to long-term and sustainable CDW management.

**Author Contributions:** Conceptualization, K.T.T., N.T.N. and K.K.; methodology, K.T.T. and K.K.; software, K.T.T. and K.K.; validation, G.H.N., N.T.N. and K.K.; formal analysis, K.T.T. and K.K.; investigation, resources, and data collection, N.T.N., K.T.T. and K.K.; writing-original draft preparation, K.T.T.; writing-review and editing, K.K.; visualization, K.T.T. and K.K.; supervision, T.I., G.H.N. and K.K.; project administration, G.H.N. and K.K.; funding acquisition, G.H.N. and K.K. All authors have read and agreed to the published version of the manuscript.

**Funding:** This research was funded by Japan Science and Technology Agency- JST and Japan International Cooperation Agency- JICA through the Science and Technology Research Partnership for Sustainable Development (SATREPS) program, grant number No. JPMJSA1701.

**Informed Consent Statement:** Not applicable.

**Data Availability Statement:** The data presented in this study are available on request from the corresponding author. The data are not publicity available due to the information security conditions of the project.

**Acknowledgments:** The authors also would like to express their gratitude to Viet Nam Solid Waste Treatment and Recycling Joint Stock Company for their cooperation in conducting this study.

**Conflicts of Interest:** The authors declare no conflict of interest.

# Appendix A

**Table A1.** Summary of legal documents, policies, and regulations on solid waste management and CDW in Hai Phong City.

| Legal Documents | Remarks |
|---|---|
| Resolution No. 09/2010/NQ-HDND dated on 15/07/2010 of the City People's Council on tasks and solutions for rural solid waste collection and treatment in the city for the period 2010–2020 Program No. 5741/CTr-UBND dated on 04/10/2010 on the implementation of Resolution No. 09/NQ-HDND of the City People's Council | Specific objectives: To improve the rural environment, by 2015: 70% of solid waste will be generated in rural areas; 70% in craft villages is collected and treated to meet environmental standards; 70% of industrial hazardous solid waste, 80% of medical hazardous solid waste, and 90% of non-hazardous industrial solid waste are collected and treated according to regulations. In 2020: 90% of solid waste will be generated in rural areas; 90% in craft villages is collected and treated to meet environmental standards; 100% of industrial solid waste and medical hazardous solid waste is collected and treated according to regulations. |
| Decision No. 1711/2012/UBND dated 11 October 2012 [37] approved the master plan on solid waste management in the city until 2025 Decision No. 1999/QD-UBND dated on 15 November 2012 approved the master plan of rural solid waste collection network in Hai Phong City until 2020. | The master plan on solid waste treatment in Hai Phong City to 2025 has the following contents:<br>- Determining generation sources and estimating the volume of solid waste in 2025 including: 1.25 million tons/year of domestic solid waste; of which 20% is construction waste (specifically, 550 tons/day in urban districts, 55 tons/day in towns, 80 tons/day in rural areas); 2290 tons/day industrial solid waste; 1940 tons/day Hazardous solid waste; 10.14 tons/day Medical solid waste (with 2.14 tons/day as hazardous waste).<br>- Assignment of responsibility for management of each type of solid waste has been clarified to each for each industry regulator.<br>- Approved a master plan for solid waste treatment zones and transit stations in Hai Phong City, including 7 city-level municipal solid waste treatment zones (Gia Minh, Tran Duong, Dong Van, Trang Cat, Dinh Vu, An Son—Lai Xuan, Quang Trung—Quang Hung), 7 district-level solid waste treatment zones (Tan Trao, Ngoc Chu, Cap Tien, Dong Bai, Ang Cha, Minh Tan, Bach Long Vy), 5 construction solid waste disposal sites, and 3 solid waste transfer stations.<br>- The solid waste treatment master plan in Hai Phong City is divided into 2 phases: (i) first stage from 2011 to 2015; (ii) second stage from 2015 to 2025. |
| Decision No. 1259/QĐ-UBND dated on 8 June 2015 | Adjustment of investment project on construction of Hai Phong City solid waste management and treatment works, invested by Hai Phong Urban Environment Company Limited: Phase 1 uses microbial incubation technology to treat urban waste; provide industrial equipment to standardize waste into useful organic products; providing garbage receiving equipment and equipment for a composting plant with a capacity of handling 200 tons of municipal waste per day; building a guard house, embankment, isolated green trees, internal roads, rainwater drainage, leachate drainage and burial plot No. 2 (area 5625 m$^2$) in the remaining land (area 39,965 m$^2$) according to the approved plan in Decision No. 1810/QD-UBND dated 10/11/2011 of the City People's Committee. The project allots 18.5 ha to build a solid waste treatment plant and landfill, with investment capital using concessional loans from the Korea Economic Development Cooperation Fund. The project is expected to be completed in 2015. |
| Decision No. 3002/QĐ-UBND dated on 1 December 2016 | Issue action plan on some urgent tasks and solutions for environmental protection in Hai Phong City. There are 14 priority actions for implementation, with the fourth being reviewing and approving according to the authority of the solid waste management planning in 2017. |

**Table A1.** *Cont.*

| Legal Documents | Remarks |
|---|---|
| Plan No. 05/2019/KH-UBND dated on 5 January 2019 Solid waste management plan to 2025, with vision toward 2050 in Hai Phong city | Key contents are as follows:<br><br>- From 90 to 100% of the total amount of waste generated from generation sources will be collected, treated, reused, or recycled to meet environmental protection requirements. Especially, 90% of construction and demolition waste generated in urban areas will be collected and treated to meet environmental protection requirements, of which 60% will be reused or recycled into recycled products and materials. Eighty percent of the ash, slag, and gypsum generated from thermal power plants, and chemical fertilizer plants will be recycled, reused and processed as raw materials for production of construction materials, levelling, etc.<br>- The plan proposed general solutions, such as completing the policy system on solid waste management, promoting the implementation of national and ministerial-level programs on research, adjusting the planning for solid waste management, establishing local solid waste databases, cooperation with other countries, international organizations, and non-governmental organizations, etc.<br>- The plan assigned the tasks for each agency in charge of solid waste management, including DONRE, DARD (Department of Agriculture and Rural Development), DOC, DOH (Department of Health), DOF (Department of Finance), DOST, DIC (Department of Information and Communications), DOET (Department of Education and Training), DHA (Department of Home Affairs), Districts People's Committee. |
| Directive No. 22/CT-UBND dated on 17 October 2019 | Directive of the City People's Committee on strengthening the sanitation, management, collection and treatment of solid waste in the city. |
| Resolution No. 09/NQ-HDND dated on 15 October 2020 of the 16th Congress of the Party Committee of Hai Phong City, term 2020–2025, | The resolution specified that the rate of daily-life solid waste collected and treated in urban areas should reach 100% and 95% in rural areas; of which over 50% will be treated by modern technologies, not landfilling, by 2025. The main task and solutions are to quickly deploy investment projects to build waste treatment plants with modern and advanced technologies to use instead of landfill and enhance value recovery from waste, strictly control all types of solid waste generated in the city in order to protect the environment, and protect human health in the future, towards the goal of sustainable development of Hai Phong City. |
| Announcement No. 293/TB-UBND dated on 23 July 2021 | Notify of the City People's Committee on the collection and treatment of solid waste in the city |
| Decision No. 2672/2021/QĐ-UBND date on 16 September 2021 | Transfer and supplement functions and tasks of solid waste management (including CDW) in the city from the DOC to the DONRE from 1 October 2021, including tasks, documents, records; facilities, equipment, assets, finance related to solid waste management functions and tasks. According to the provisions of Decree No. 107/2020/ND-CP dated 14 September 2020 of the Government and Circular No. 05/2021/TT-BTNMT dated on 29 May 2021 of the MONRE. |
| Plan No. 212/KH-UBND dated on 17 September/2021 | Plan on plastic waste management in the 2021–2026 period in Hai Phong City. |
| Decision No. 2799/2021/QĐ-UBND dated on 28 September 2021 | Approval of the report on the current state of the environment in Hai Phong City for the period 2016–2020. |

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
