# Peer review of "Management Assessment and Future Projections of Construction and Demolition Waste Generation in Hai Phong City, Vietnam"

_sustainability, doi:10.3390/su14159628_

Round 1
Reviewer 1 Report
Dear authors,
the study you developed could be of useful for a future better management of CDW in Vietnam.
Some comments for enhancing your work:
1. The introduction is very long and has a lot of numbers, it would be easier to read a more concise introduction.
2. Some concepts are repeted: line 91-92 says the same conceptof line 112; lines 105 and 107 are very similar. Please, organise the introduction avoiding repetitions.
3. The aim of the study arrives just at line 149, I suggest you to introduce the goal of your study before.
4. Line 224: CC-CB seems CC minus CB, while in your formula is exactly the opposite. It would be clearer to say CC+CB.
5. The novelty of the study is probably limited. However, the information you gathered could be very useful for the CDW management in Vietnam. For this reason, I suggest you to include a spreadsheet file as Supplementary Material, where the consultation of data and the use of your formula to predict the amount of CDW result clearer and more direct.
Author Response
Response to Reviewer 1 Comments
Point 1: The introduction is very long and has a lot of numbers, it would be easier to read a more concise introduction.
Response 1: Following your recommendation, we have revised the introduction to make it more concise.
Point 2: Some concepts are repeted: line 91-92 says the same conceptof line 112; lines 105 and 107 are very similar. Please, organise the introduction avoiding repetitions
Response 2: Similar to the previous comment, we have considered your suggestions and reorganise the introduction avoiding repetitions
Point 3: The aim of the study arrives just at line 149, I suggest you to introduce the goal of your study before
Response 3: Similar to the previous comment, we have introduced the study goal following your suggestion.
Point 4: Line 224: CC-CB seems CC minus CB, while in your formula is exactly the opposite. It would be clearer to say CC+CB
Response 4: Following your advice, we have changed “CC-CB” to “CC+CB” to avoid confusion.
Point 5: The novelty of the study is probably limited. However, the information you gathered could be very useful for the CDW management in Vietnam. For this reason, I suggest you to include a spreadsheet file as Supplementary Material, where the consultation of data and the use of your formula to predict the amount of CDW result clearer and more direct.
Response 5: We appreciate your advice and have incorporated your suggestions by clearing the section Methodologies related to our study’s objectives and methods. We also added a spreadsheet file as Supplementary Material which including the consultation of data and formula to predict the amount of CDW results.
Reviewer 2 Report
The paper presents the results of field research (interview type) on the construction and demolition waste generated in Hai Phong City, Vietnam. Data collected in the field were used in models for future projections.
For the text to be published, some adjustments must be made:
1) The title does not represent the study's objective (from the title, the study seems to be a literature review - not an interview for data collection with subsequent modeling/projection).
2) improve the description of the method in the abstract.
3) In the introduction, better describe the problem of the study.
4) Table 2 is very confusing. Improve presentation. Same for table 5.
5) Figure 1 is not legible. Improve the quality. Same for figure 9.
6) Deepen the discussion of the results, including more correlations with the literature.
7) How to reduce illegal CDW deposits?
8) Rewrite the conclusions - including the main results of the study.
9) References can be extended.
Author Response
Response to Reviewer 2 Comments
Point 1: The title does not represent the study's objective (from the title, the study seems to be a literature review - not an interview for data collection with subsequent modeling/projection)..
Response 1: Following your advice, we have chaged the title “Current Management and Future Projections of Construction and Demolition Waste Generation in Hai Phong City, Vietnam to “Management Assessment and Future Projections of Construction and Demolition Waste Generation in Hai Phong City, Vietnam”.
Point 2: Improve the description of the method in the abstract.
Response 2: We appreciate your advice and have revised the abstract accordingly.
Point 3: In the introduction, better describe the problem of the study.
Response 3: Similar to the previous comment, we have introduced the problem of the study in the revised introduction.
Point 4: Table 2 is very confusing. Improve presentation. Same for table 5.
Response 4: Table 2 is the information of the subjects and contents in the survey forms. Table 5 is the result of gathering survey information at official and illegal CDW dumps. Thus, we like to keep as it.
Point 5: Figure 1 is not legible. Improve the quality. Same for figure 9.
Response 5: Figure 1 and Figure 9 have been revised for more clearify
Point 6: Deepen the discussion of the results, including more correlations with the literature
Response 6: We appreciate your advice and have revised the discussions accordingly.
Point 7: How to reduce illegal CDW deposits?
Response 7: It is a big challenge to control the illegal CDW dumping. Base on the our survey data and future projections shown in this manuscript, the authors will continuously discuss the countermeasures toghether with central and local authoritries, and all stakeholders.
Point 8: Rewrite the conclusions - including the main results of the study.
Response 8: Following your recommendation, we have revised the conclusions.
Point 9: References can be extended.
Response 9: We appreciate your advice and We have only cited references that are directly related to the results and content of the paper.
Reviewer 3 Report
The article presents a very relevant theme for the scientific community. The authors propose a model to forecast future generation of CDW by combining the investigated data and a multivariable regression analysis that can contribute to the sustainable CDW management.
The paper is very well structured and well written. The introduction is extensive and up to date. Figures and tables are clear and objective. Therefore, I have no contributions to do for this manuscript.
Author Response
Response to Reviewer 3 Comments
Point 1: The paper is very well structured and well written. The introduction is extensive and up to date. Figures and tables are clear and objective. Therefore, I have no contributions to do for this manuscript..
Response 1: We appreciate your comments and have revised some content for enhancing the manuscrip quality.
Round 2
Reviewer 1 Report
Dear authors,
Thank you for your revised paper, which has improved.
I'm not sure that the title is very effective... Is correct in english "Management assessment"?
Reviewer 2 Report
The authors carried out the suggested revisions. I recommend the paper's approval.